# Simulating Action Dynamics with Neural Process Networks

**Antoine Bosselut**[†]**, Omer Levy**[†]**, Ari Holtzman**[†]**, Corin Ennis**[‡]**, Dieter Fox**[†] **& Yejin Choi**[†]
[†]Paul G. Allen School of Computer Science & Engineering
University of Washington
{antoineb,omerlevy,ahai,fox,yejin}@cs.washington.edu
[‡]School of Science, Technology, Engineering & Mathematics
University of Washington - Bothell
{corin123}@uw.edu

## Abstract

Understanding procedural language requires anticipating the causal effects of actions, even when they are not explicitly stated. In this work, we introduce Neural Process Networks to understand procedural text through (neural) simulation of action dynamics. Our model complements existing memory architectures with dynamic entity tracking by explicitly modeling actions as state transformers. The model updates the states of the entities by executing learned action operators. Empirical results demonstrate that our proposed model can reason about the unstated causal effects of actions, allowing it to provide more accurate contextual information for understanding and generating procedural text, all while offering more interpretable internal representations than existing alternatives.

## 1 Introduction

Understanding procedural text such as instructions or stories requires anticipating the implicit causal effects of actions on entities. For example, given instructions such as "*add* blueberries to the muffin mix, then *bake* for one half hour," an intelligent agent must be able to anticipate a number of entailed facts (e.g., the blueberries are now in the oven; their "temperature" will increase). While this common sense reasoning is trivial for humans, most natural language understanding algorithms do not have the capacity to reason about causal effects not mentioned directly in the surface strings (Levy et al., 2015; Jia & Liang, 2017; Lucy & Gauthier, 2017).

In this paper, we introduce Neural Process Networks, a procedural language understanding system that tracks common sense attributes through neural simulation of action dynamics. Our network models interpretation of natural language instructions as a *process* of actions and their cumulative effects on entities. More concretely, reading one sentence at a time, our model attentively selects what actions to execute on which entities, and remembers the state changes induced with a recurrent memory structure. In Figure 1, for example, our model indexes the "tomato" embedding, selects the "wash" and "cut" functions and performs a computation that changes the "tomato" embedding so that it can reason about attributes such as its "SHAPE" and "CLEANLINESS".

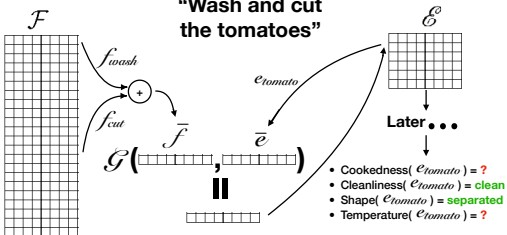

Figure 1: The *process* is a narrative of entity state changes induced by actions. In each sentence, these state changes are induced by simulated actions and must be remembered.

Our model contributes to a recent line of research that aims to model aspects of world state changes, such as language models and machine readers with explicit entity representations (Henaff et al., 2016; Yang et al., 2016; Ji et al., 2017), as well as other more general purpose memory network

variants (Weston et al., 2014; Sukhbaatar et al., 2015; Hill et al., 2015; Seo et al., 2016). This *world-centric* modeling of procedural language (i.e., understanding by simulation) abstracts away from the surface strings, complementing *text-centric* modeling of language, which focuses on syntactic and semantic labeling of surface words (i.e., understanding by labeling).

Unlike previous approaches, however, our model also learns explicit action representations as functional operators (See Figure 1). While representations of action semantics could be acquired through an embodied agent that can see and interact with the world (Oh et al., 2015), we propose to learn these representations from text. In particular, we require the model to be able to *explain* the causal effects of actions by predicting natural language attributes about entities such as "LOCATION" and "TEMPERATURE". The model adjusts its representations of actions based on errors it makes in predicting the resultant state changes to attributes. This textual simulation allows us to model aspects of action causality that are not readily available in existing simulation environments. Indeed, most virtual environments offer limited aspects of the world – with a primary focus on spatial relations (Oh et al., 2015; Chiappa et al., 2017; Wahlstrom et al., 2015). They leave out various other dimensions of the world states that are implied by diverse everyday actions such as "dissolve" (change of "COMPOSITION") and "wash" (change of "CLEANLINESS").

Empirical results demonstrate that parametrizing explicit action embeddings provides an inductive bias that allows the neural process network to learn more informative context representations for understanding and generating natural language procedural text. In addition, our model offers more interpretable internal representations and can reason about the unstated causal effects of actions explained through natural language descriptors. Finally, we include a new dataset with fine-grained annotations on state changes, to be shared publicly, to encourage future research in this direction.

## 2    NEURAL PROCESS NETWORK

The neural process network is an interpreter that reads in natural language sentences, one at a time, and simulates the process of actions being applied to relevant entities through learned representations of actions and entities.

### 2.1    OVERVIEW AND NOTATION

The main component of the neural process network is the simulation module (§2.5), a recurrent unit whose internals simulate the effects of actions being applied to entities. A set of $V$ actions is known a priori and an embedding is initialized for each one, $\mathcal{F} = \{f_1, ...f_V\}$. Similarly, a set of $I$ entities is known and an embedding is initialized for each one: $\mathcal{E} = \{e_1, ...e_I\}$. Each $e_i$ can be considered to encode information about state attributes of that entity, which can be extracted by a set of state predictors (§2.6). As the model reads text, it "applies" action embeddings to the entity vectors, thereby changing the state information encoded about the entities.

For any document $d$, an initial list of entities $I_d$ is known and $\mathcal{E}_d = \{e_i | i \in I_d\} \subset \mathcal{E}$ entity state embeddings are initialized. As the neural process network reads a sentence from the document, it selects a subset of both $\mathcal{F}$ (§2.3) and $\mathcal{E}_d$ (§2.4) based on the actions performed and entities affected in the sentence. The entity state embeddings are changed by the action and the new embeddings are used to predict end states for a set of state changes (§2.6). The prediction error for end states is backpropagated to the action embeddings, learning action representations that model the simulation of desired causal effects on entities. This process is broken down into five modules below. Unless explicitly defined, all $W$ and $b$ variables are parametrized linear projections and biases. We use the notation $\{e_i\}_t$ when referring to the values of the entity embeddings before processing sentence $s_t$.

### 2.2    SENTENCE ENCODER

Given a sentence $s_t$, a Gated Recurrent Unit (Cho et al., 2014) encodes each word and outputs its last hidden vector as a sentence encoding $h_t$ (Sutskever et al., 2014).

### 2.3    ACTION SELECTOR

Given $h_t$ from the sentence encoder, the action selector (bottom left in Fig. 2) contextually determines which action(s) from $\mathcal{F}$ to execute. For example, if the input sentence is *"wash and cut*

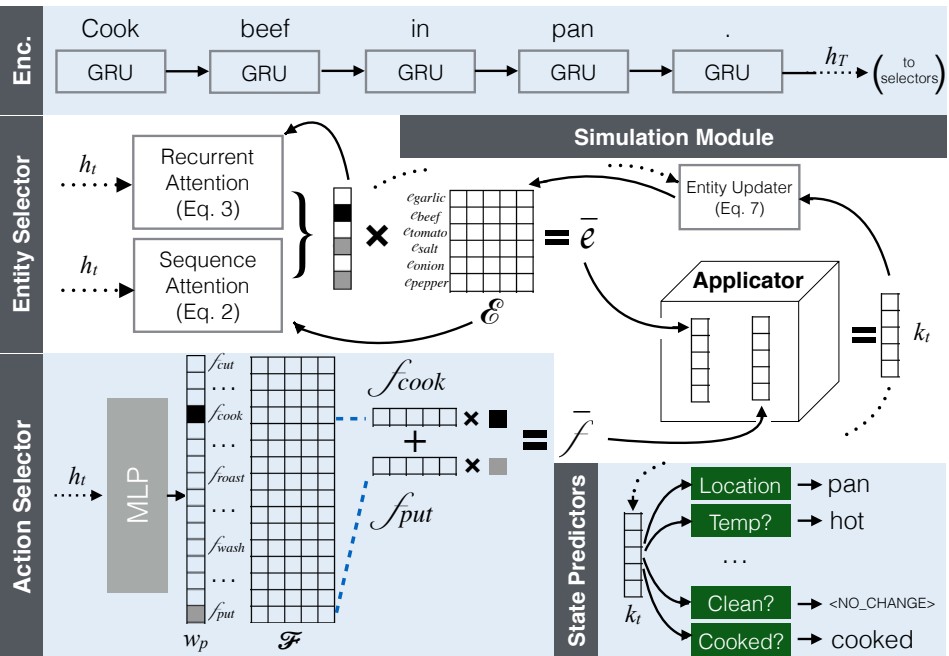

Figure 2: Model Summary. The sentence encoder converts a sentence to a vector representation, $h_t$. The action selector and entity selector use the vector representation to choose the actions that are applied and the entities that are acted upon in the sentence. The simulation module indexes the action and entity state embeddings, and applies the transformation to the entities. The state predictors predict the new state of the entities if a state change has occurred. Equation references are provided in parentheses.

*beets"*, both $f_{wash}$ and $f_{cut}$ must be selected. To account for multiple actions, we make a soft selection over $\mathcal{F}$, yielding a weighted sum of the selected action embeddings $\bar{f}_t$:

$$w_p = \text{MLP}(h_t)$$
$$\bar{w}_p = \frac{w_p}{\sum_{j=1}^{V} w_{p_j}} \tag{1}$$
$$\bar{f}_t = \bar{w}_p^\intercal \mathcal{F}$$

where MLP is a parametrized feed-forward network with a sigmoid activation and $w_p \in \mathbb{R}^V$ is the attention distribution over $V$ possible actions (§3.1). We compose the action embedding by taking the weighted average of the selected actions.

## 2.4 ENTITY SELECTOR

**Sentence Attention** Given $h_t$ from the sentence encoder, the entity selector chooses relevant entities using a soft attention mechanism:

$$\tilde{h}_t = \text{ReLU}(W_1 h_t + b_1)$$
$$d_i = \sigma(e_{i_0}^\intercal W_2 [\tilde{h}_t; w_p]) \tag{2}$$

where $W_2$ is a bilinear mapping, $e_{i_0}$ is a unique key for each entity (§2.5), and $d_i$ is the attention weight for entity embedding $e_i$. For example, in *"wash and cut beets and carrots"*, the model should select $e_{beet}$ and $e_{carrot}$.

**Recurrent Attention** While sentence attention would suffice if entities were always explicitly mentioned, natural language often elides arguments or uses referent pronouns. As such, the module must be able to consider entities mentioned in previous sentences. Using $\tilde{h}_t$, the model computes a soft choice over whether to choose affected entities from this step's attention $d_i$ or the previous step's attention distribution.

$$c = softmax(W_3 \tilde{h}_t + b_3)$$
$$a_{i_t} = c_1 d_i + c_2 a_{i_{t-1}} + c_3 \mathbf{0} \tag{3}$$

where $c \in \mathbb{R}^3$ is the choice distribution, $a_{i_{t-1}}$ is the previous sentence's attention weight for each entity, $a_{i_t}$ is the final attention for each entity, and $\mathbf{0}$ is a vector of zeroes (providing the option to not change any entity). Prior entity attentions can propagate forward for multiple steps.

## 2.5 SIMULATION MODULE

**Entity Memory**   A unique state embedding $e_i$ is initialized for every entity $i$ in the document. A unique key to index each embedding $e_{i_0}$ is set as the initial value of the embedding (Henaff et al., 2016; Miller et al., 2016). After the model reads $s_t$, it modifies $\{e_i\}_t$ to reflect changes influenced by actions. At every time step, the entity memory receives the attention weights from the entity selector, normalizes them and computes a weighted average of the relevant entity state embeddings:

$$\alpha_i = \frac{a_i}{\sum_{j=1}^{I_d} a_j} \qquad (4) \qquad\qquad \bar{e}_t = \sum_{j=1}^{I_d} \alpha_i e_i \qquad (5)$$

**Applicator**   Given the action summary embedding $\bar{f}_t$ and the entity summary embedding $\bar{e}_t$, the applicator (middle right in Fig. 2) applies the selected actions to the selected entities, and outputs the new proposal entity embedding $k_t$.

$$k_t = \text{ReLU}(\bar{f}_t W_4 \bar{e}_t + b_4) \qquad (6)$$

where $W_4$ is a third order tensor projection. The vector $k_t$ is the new representation of the entity $\bar{e}_t$ after the applicator simulates the action being applied to it.

**Entity Updater**   The entity updater interpolates the new proposal entity embedding $k_t$ and the set of current entity embeddings $\{e_i\}_t$:

$$e_{i_{t+1}} = a_{i_t} k_t + (1 - a_{i_t}) e_{i_t} \qquad (7)$$

yielding an updated set of entity embeddings $\{e_i\}_{t+1}$. Each embedding is updated proportional to its entity's unnormalized attention $a_i$, allowing the model to completely overwrite the state embedding for any entity. For example, in the sentence *"mix the flour and water,"* the embeddings for $e_{flour}$ and $e_{water}$ must both be overwritten by $k_t$ because they no longer exist outside of this new composition.

## 2.6 STATE PREDICTORS

Given the new proposal entity embedding $k_t$, the state predictor (bottom right in Fig. 2) predicts changes to the resulting entity embedding $k_t$ along the following six dimensions: location, cookedness, temperature, composition, shape, and cleanliness. Discrete multi-class classifiers, one for each dimension, take in $k_t$ and predict a unique end state for their corresponding state change type:

$$P(Y_s | k_t) = softmax(W_s k_t + b_s) \qquad (8)$$

For location changes, which require contextual information to predict the end state, $k_t$ is concatenated with the original sentence representation $h_t$ to predict the final state.

## 3 TRAINING

### 3.1 STATE CHANGE KNOWLEDGE

In this work we focus on physical action verbs in cooking recipes. We manually collect a set of 384 actions such as *cut, bake, boil, arrange*, and *place*, organizing their causal effects along the following predefined dimensions: LOCATION, COOKEDNESS, TEMPERATURE, SHAPE, CLEANLINESS and COMPOSITION. The textual simulation operated by the model induces state changes along these dimensions by applying actions functions from the above set of 384. For example, *cut* entails a change in SHAPE, while *bake* entails a change in TEMPERATURE, COOKEDNESS, and even LOCATION. We annotate the state changes each action induces, as well as the end state of the action, using Amazon Mechanical Turk. The set of possible end states for a state change can range from 2 for binary state changes to more than 200 (See Appendix C for details). Table 1 provides examples of annotations in this action lexicon.

| Action | State Change Types | End States |
|--------|-------------------|------------|
| braise | COOKEDNESS; TEMPERATURE | COOKED; HOT |
| chill | TEMPERATURE | COLD |
| knead | SHAPE | MOLDED |
| wash | CLEANLINESS | CLEAN |
| dissolve | COMPOSITION | COMPOSED |
| refrigerate | TEMPERATURE; LOCATION | COLD; REFRIGERATOR |
| slice | SHAPE | SEPARATED |

Table 1: Example actions, the state changes they induce, and the possible end states

## 3.2 DATASET

For learning and evaluation, we use a subset of the Now You're Cooking dataset (Kiddon et al., 2016). We chose 65816 recipes for training, 175 recipes for development, and 700 recipes for testing. For the development and test sets, crowdsourced workers densely annotate actions, entities and state changes that occur in each sentence so that we can tune hyperparameters and evaluate on gold evaluation sets. Annotation details are provided in Appendix C.3.

## 3.3 COMPONENT-WISE TRAINING

The neural process network is trained by jointly optimizing multiple losses for the action selector, entity selector, and state change predictors. Importantly, our training scheme uses weak supervision because dense annotations are prohibitively expensive to acquire at a very large scale. Thus, we heuristically extract verb mentions from each recipe step and assign a state change label based on the state changes induced by that action (§3.1). Entities are extracted similarly based on string matching between the instructions and the ingredient list. We use the following losses for training:

**Action Selection Loss** Using noisy supervision, the action selector is trained to minimize the cross-entropy loss for each possible action, allowing multiple actions to be chosen at each step if multiple actions are mentioned in a sentence. The MLP in the action selector (Eq. 1) is pretrained.

**Entity Selection Loss** Similarly, to train the attentive entity selector, we minimize the binary cross-entropy loss of predicting whether each entity is affected in the sentence.

**State Change Loss** For each state change predictor, we minimize the negative loglikelihood of predicting the correct end state for each state change.

**Coverage Loss** An underlying assumption in many narratives is that all entities that are mentioned should be important to the narrative. We add a loss term that penalizes narratives whose combined attention weights for each entity does not sum to more than 1.

$$\mathcal{L}_{cover} = -\frac{1}{I_d} \sum_{i=1}^{I_d} \log \sum_{t=1}^{S} a_{i_t} \tag{9}$$

where $a_{i_t}$ is the attention weight for a particular entity at sentence $t$ and $I_d$ is the number of entities in a document. $\sum_{t=1}^{S} a_{i_t}$ is upper bounded by 1. This is similar to the coverage penalty used in neural machine translation (Tu et al., 2016).

## 4 EXPERIMENTAL SETUP

We evaluate our model on a set of intrinsic tasks centered around tracking entities and state changes in recipes to show that the model can simulate preliminary dynamics of the recipe task. Additionally, we provide a qualitative analysis of the internal components of our model. Finally, we evaluate the quality of the states encoded by our model on the extrinsic task of generating future steps in a recipe.

### 4.1 INTRINSIC EVALUATION - TRACKING

In the tracking task, we evaluate the model's ability to identify which entities are selected and what changes have been made to them in every step. We break the tracking task into two separate evalua-

| Model | Entity Selection | | | State Change | |
|---|---|---|---|---|---|
| | F1 | UR | CR | F1 | ACC |
| 2-layer LSTM Entity Recognizer | 50.98 | 74.03 | 13.33 | - | - |
| Adapted Gated Recurrent Unit | 45.94 | 67.69 | 7.74 | 41.16 | 52.69 |
| Adapted Recurrent Entity Network | 48.57 | 71.88 | 9.87 | 42.32 | 53.47 |
| - Recurrent Attention (Eq. 3) | 48.91 | 72.32 | 12.67 | 42.14 | 50.48 |
| - Coverage Loss (Eq. 9) | 55.18 | 73.98 | 20.23 | 44.44 | **55.20** |
| - Action Connections (Eq. 2) | 54.85 | 73.54 | 20.03 | 44.05 | 54.81 |
| - Action Selector Pretraining | 54.91 | 73.87 | 20.30 | 44.28 | 55.00 |
| + Pretrained Action Embeddings | 55.16 | 74.02 | 20.32 | 44.02 | 55.03 |
| - Action Embedding Updates | 53.79 | 70.77 | 18.60 | 44.27 | 55.02 |
| Full Model | **55.39** | **74.88** | **20.45** | **44.65** | 55.07 |

Table 2: Results for entity selection and state change selection

tions, entity selection and end state prediction, and also investigate whether the model learns internal representations that approximate recipe dynamics.

**Metrics** In the entity selection test, we report the F1 score of choosing the correct entities in any step. A selected entity is defined as one whose attention weight $a_i$ is greater than 50% (§2.4). Because entities may be harder to predict when they have been combined with other entities (e.g., the mixture may have a new name), we also report the recall for selecting combined (CR) and uncombined (UR) entities. In the end state prediction test, we report how often the model correctly predicts the state change performed in a recipe step and the resultant end state. This score is then scaled by the accuracy of predicting which entities were changed in that same step. We report the average F1 and accuracy across the six state change types.

**Baselines** We compare our models against two baselines. First, we built a GRU model that is trained to predict entities and state changes independently. This can be viewed as a bare minimum network with no action representations or recurrent entity memory. The second baseline is a Recurrent Entity Network (Henaff et al., 2016) with changes to fit our task. First, the model can tie memory cells to a subset of the full list of entities so that it only considers entities that are present in a particular recipe. Second, the entity distribution for writing to the memory cells is re-used when we query the memory cells. The normalized weighted average of the entity cells is used as the input to the state predictors. The unnormalized attention when writing to each cell is used to predict selected entities. Both baselines are trained with entity selection and state change losses (§3.3).

**Ablations** We report results on six ablations. First, we remove the recurrent attention (Eq. 3). The model only predicts entities using the current encoder hidden state. In the second ablation, the model is trained with no coverage penalty (Eq. 9). The third ablation prunes the connection from the action selector $w_p$ to the entity selector (Eq. 2). We also explore not pretraining the action selector. Finally, we look at two ablations where we intialize the action embeddings with vectors from a skipgram model. In the first, the model operates normally, and in the second, we do not allow gradients to backpropagate to the action embeddings, updating only the mapping tensor $W_4$ instead (Eq. 6).

## 4.2 EXTRINSIC EVALUATION - GENERATION

The generation task tests whether our system can produce the next step in a recipe based on the previous steps that have been performed. The model is provided all of the previous steps as context.

**Metrics** We report the combined BLEU score and ROUGE score of the generated sequence relative to the reference sequence. Each candidate sequence has one reference sentence. Both metrics are computed at the corpus-level. Also reported are "VF1", the F1 score for the overlap of the actions performed in the reference sequence and the verbs mentioned in the generated sequence, and "SF1", the F1 score for the overlap of end states annotated in the reference sequence and predicted by the generated sequences. End states for the generated sequences are extracted using the lexicon from Section 3.1 based on the actions performed in the sentence.

**Setup** To apply our model to the task of recipe step generation, we input the context sentences through the neural process network and record the entity state vectors once the entire context has

| | | |
|---|---|---|
| Good | $s_{t-1}$ | Add tomato paste, broth, garlic, chili powder, cumin, chile peppers, and water. |
| | $s_t$ | Bring to boil, then turn very low, cover and simmer until meat is tender. |
| | **selected** | meat, garlic, chili powder, tomato paste, cumin, chiles, beef broth, water |
| | **correct** | Same + [oil] |
| Good | $s_{t-4}$ | Stir in oats, sugar, flour, corn syrup, milk, vanilla extract, and salt. |
| | $s_{t-3}$ | Mix well. |
| | $s_{t-2}$ | Drop by measuring teaspoonfuls onto cookie sheets. |
| | $s_{t-1}$ | Bake 5 - 7 minutes. |
| | $s_t$ | Let cool. |
| | **selected** | oats, sugar, flour, corn syrup, milk, vanilla extract, salt |
| | **correct** | oats, sugar, flour, corn syrup, milk, vanilla extract, salt |
| Good | $s_{t-1}$ | In a large saucepan over low heat, melt marshmallows. |
| | $s_t$ | Add sprinkles, cereal, and raisins, stir until well coated. |
| | **selected** | marshmallows, cereal, raisins |
| | **correct** | marshmallows, cereal, raisins, sprinkles |
| Bad | $s_{t-3}$ | Ladle the barbecue sauce around the crust and spread. |
| | $s_{t-2}$ | Add mozzarella, yellow cheddar, and monterey jack cheese. |
| | $s_{t-1}$ | Next, add onion mixture and sliced chicken breast . |
| | $s_t$ | Top pizza with jalapeno peppers. |
| | **selected** | jalapenos |
| | **correct** | crust, sauce, mozzarella, cheddar, monterey jack, white onion, chicken, jalapenos |
| Bad | $s_{t-2}$ | Combine 1 cup flour, salt, and 1 tbsp sugar. |
| | $s_{t-1}$ | Cut in butter until mixture is crumbly, then sprinkle with vinegar . |
| | $s_t$ | Gather dough into a ball and press into bottom of 9 inch springform pan. |
| | **selected** | butter, vinegar |
| | **correct** | flour, salt, sugar, butter, vinegar |

Table 3: Examples of the model selecting entities for sentence $s_t$. The previous sentences are provided as context in cases where they are relevant.

been read (§2.5). These vectors can be viewed as a snapshot of the current state of the entities once the preceding context has been simulated inside the neural process network. We encode these vectors using a bidirectional GRU (Cho et al., 2014) and take the final time step hidden state $e_I$. A different GRU encodes the context words in the same way (yielding $h_T$) and the first hidden state input to the decoder is computed using the projection function:

$$\tilde{h}_0 = W_5(e_I \circ h_T) \tag{10}$$

where $\circ$ is the Hadamard product between the two encoder outputs. All models are trained by minimizing the negative loglikelihood of predicting the next word for the full sequence. Implementation details can be found in Appendix A.

**Baselines** For the generation task, we use three baselines: a seq2seq model with no attention (Sutskever et al., 2014), an attentive seq2seq model (Bahdanau et al., 2014), and a similar variant as our NPN generator, except where the entity states have been computed by the Recurrent Entity Network (EntNet) baseline (§4.1). Implementation details for baselines can be found in Appendix B.

## 5 EXPERIMENTAL RESULTS

### 5.1 INTRINSIC EVALUATIONS

**Entity Selection** As shown in Table 8, our full model outperforms all baselines at selecting entities, with an F1 score of 55.39%. The ablation study shows that the recurrent attention, coverage loss, action connections and action selector pretraining improve performance. Our success at predicting entities extends to both uncomposed entities, which are still in their raw forms (e.g., melt the *butter* → butter), and composed entities, in which all of the entities that make up a composition must be selected. For example, in a *Cooking lasagna* recipe, if the final step involves baking the prepared lasagna, the model must select all the entities that make up the lasagna (e.g., lasagna sheets, beef, tomato sauce). In Table 3, we provide examples of our model's ability to handle complex cases such

| Action | Nearest Neighbor Actions |
|---|---|
| cut | slice, split, snap, slash, carve, slit, chop |
| boil | cook, microwave, fry, steam, simmer |
| add | sprinkle, mix, reduce, splash, stir, dust |
| wash | rinse, scrub, refresh, soak, wipe, scale |
| mash | spread, puree, squeeze, liquefy, blend |
| place | ease, put, lace, arrange, leave |
| rinse | wash, refresh, soak, wipe, scrub, clean |
| warm | reheat, ignite, heat, light, crisp, preheat |
| steam | microwave, crisp, boil, parboil, heat |
| sprinkle | top, pat, add, dip, salt, season |
| grease | coat, rub, dribble, spray, smear, line |

Table 4: Most similar actions based on cosine similarity of action embeddings

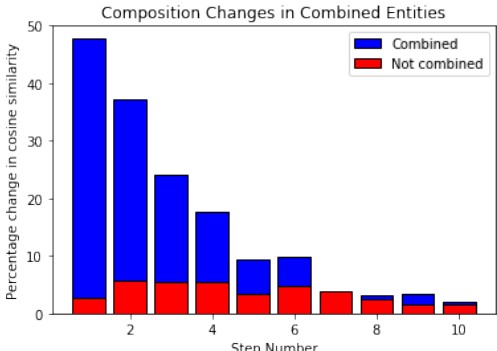

Figure 3: Change in cosine similarity of entity state embeddings

| Model | BLEU | ROUGE-L | VF1 | SF1 |
|---|---|---|---|---|
| Vanilla Seq2Seq | 2.81 | 33.00 | 16.17 | 40.21 |
| Attentive Seq2Seq | 2.83 | 33.18 | 16.97 | 41.43 |
| EntNet Generator | 2.30 | 32.71 | 17.53 | 42.43 |
| NPN Generator | **3.74** | **35.64** | **20.12** | **43.40** |

Table 5: Generation Results

as compositional entities (Ex. 1, 3), and elided arguments over long time windows (Ex. 2). We also provide examples where the model fails to select the correct entities because it does not identify the mapping between a reference construct such as "pizza" (Ex. 4) or "dough" (Ex. 5) and the set of entities that composes it, showcasing the difficulty of selecting the full set for a composed entity.

**State Change Tracking**  In Table 8, we show that our full model outperforms competitive baselines such as Recurrent Entity Networks (Henaff et al., 2016) and jointly trained GRUs. While the ablation without the coverage loss shows higher accuracy, we attribute this to the fact that it predicts a smaller number of total state changes. Interestingly, initializing action embeddings with skipgram vectors and locking their values shows relatively high performance, indicating the potential gains in using powerful pretrained representations to represent actions.

**Action Embeddings**  In our model, each action is assigned its own embedding, but many actions induce similar changes in the physical world (e.g.,"cut" and "slice"). After training, we compute the pairwise cosine similarity between each pair of action embeddings. In Table 4, we see that actions that perform similar functions are neighbors in embedding space, indicating the model has captured certain semantic properties of these actions. Learning action representations through the state changes they induce has allowed the model to cluster actions by their transformation functions.

**Entity Compositions**  When individual entities are combined into new constructs, our model averages their state embeddings (Eq. 5), applies an action embedding to them (Eq. 6), and writes them to memory (Eq. 7). The state embeddings of entities that are combined should be overwritten by the same new embedding. In Figure 3, we present the percentage increase in cosine similarity for state embeddings of entities that are combined in a sentence (blue) as opposed to the percentage increase for those that are not (red bars). While the soft attention mechanism for entity selection allows similarities to leak between entity embeddings, our system is generally able to model the compositionality patterns that result from entities being combined into new constructs.

## 5.2 EXTRINSIC EVALUATIONS

**Recipe Step Generation**  Our results in Table 5 indicate that sequences generated using the neural process network entity states as additional input yield higher scores than competitive baselines. The entity states allow the model to predict next steps conditioned on a representation of the world being simulated by the neural process network. Additionally, the higher VF1 and SF1 scores indicate that the model is indeed using the extra information to better predict the actions that should follow the context provided. Example generations for each baselines from the dev set are provided in Table 6,

| | | |
|---|---|---|
| **Context** | Preheat oven to 425 degrees. | |
| **Reference** | Melt butter in saucepan and mix in bourbon, thyme, pepper, and salt. | |
| **NPN** | Melt butter in skillet. | |
| **Seq2seq** | Lightly grease 4 x 8 baking pan with sunflower oil. | |
| **Attentive Seq2seq** | Combine all ingredients and mix well. | |
| **EntNet** | In a large bowl, combine flour, baking powder, baking soda, salt, and pepper. | |
| **Context** | Pour egg mixture over caramelized sugar in cake pan. Place cake pan in large shallow baking dish. Bake for 55 minutes or until knife inserted into flan comes out clean. | |
| **Reference** | Cover and chill at least 8 hours. | |
| **NPN** | Refrigerate until ready to use. | |
| **Seq2seq** | Serve at room temperature. | |
| **Attentive Seq2seq** | Store in an airtight container. | |
| **EntNet** | Store in an airtight container. | |
| **Context** | Cut squash into large pieces and steam. Remove cooked squash from shells; | |
| **Reference** | Measure 4 cups pulp and reserve remainder for another dish. | |
| **NPN** | Drain. | |
| **Seq2seq** | Mash pulp with a fork. | |
| **Attentive Seq2seq** | Set aside. | |
| **EntNet** | Set aside. | |

Table 6: Examples of the model generating sentences compared to baselines. The context and reference are provided first, followed by our model's generation and then the baseline generations

showing that the NPN generator can use information about ingredient states to reason about the most likely next step. The first and second examples are interesting as it shows that the NPN-aware model has learned to condition on entity state – knowing that raw butter will likely be melted or that a cooked flan must be refrigerated. The third example is also interesting because the model learns that cooked vegetables such as squash will sometimes be drained, even if it is not relevant to this recipe because the squash is steamed. The seq2seq and EntNet baselines, meanwhile, output reasonable sentences given the immediate context, but do not exhibit understanding of global patterns.

## 6 RELATED WORK

Recent studies in machine comprehension have used a neural memory component to store a running representation of processed text (Weston et al., 2014; Sukhbaatar et al., 2015; Hill et al., 2015; Seo et al., 2016). While these approaches map text to memory vectors using standard neural encoder approaches, our model, in contrast, directly interprets text in terms of the effects actions induce in entities, providing an inductive bias for learning how to represent stored memories. More recent work in machine comprehension also sought to couple the memory representation with tracking entity states (Henaff et al., 2016). Our work seeks to provide a relatively more structured representation of domain-specific action knowledge to provide an inductive bias to the reasoning process.

Neural Programmers (Neelakantan et al., 2015; 2016) have also used functions to simulate reasoning, by building a model to select rows in a database and applying operation on those selected rows. While their work explicitly defined the effect of a number of operations for those rows, we provide a framework for learning representations for a more expansive set of actions, allowing the model to learn representations for how actions change the state space.

Works on instructional language studied the task of building discrete graph representations of recipes using probabilistic models (Kiddon et al., 2015; Mori et al., 2014; 2012). We propose a complementary new model by integrating action and entity relations into the neural network architecture and also address the additional challenge of tracking the state changes of the entities.

Additional work in tracking states with visual or multimodal context has focused on 1) building graph representations for how entities change in goal-oriented domains (Gao et al., 2016; Liu et al., 2016; Si et al., 2011) or 2) tracking visual state changes based on decisions taken by agents in environment simulators such as videos or games (Chiappa et al., 2017; Wahlstrom et al., 2015; Oh et al., 2015). Our work, in contrast, models state changes in embedding space using only text-based signals to map real-world actions to algebraic transformations.

## 7 CONCLUSION

We introduced the *Neural Process Network* for modeling a process of actions and their causal effects on entities by learning action transformations that change entity state representations. The model maintains a recurrent memory structure to track entity states and is trained to predict the state changes that entities undergo. Empirical results demonstrate that our model can learn the causal effects of action semantics in the cooking domain and track the dynamic state changes of entities, showing advantages over competitive baselines.

## ACKNOWLEDGMENTS

We thank Yonatan Bisk, Peter Clark, Bhavana Dalvi, Niket Tandon, and Yuyin Sun for helpful discussions at various stages of this work. This research was supported in part by NSF (IIS-1524371, IIS-1714566, NRI-1525251), DARPA under the CwC program through the ARO (W911NF-15-1-0543), Samsung Research, and gifts by Google and Facebook.

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

## A   TRAINING DETAILS OF OUR FULL MODEL AND ABLATIONS

### A.1   TRACKING MODELS

The hidden size of the instruction encoder is 100, the embedding sizes of action functions and entities are 30. We use dropout with a rate of 0.3 before any non-recurrent fully connected layers Srivastava et al. (2014). We use the Adam optimizer (Kingma & Ba, 2014) with a learning rate of .001 and decay by a factor of 0.1 if we see no improvement on validation loss over three epochs. We stop training early if the development loss does not decrease for five epochs. The batch size is 64. We use two instruction encoders, one for the entity selector, and one for the action selector. Word embeddings and entity embeddings are initialized with skipgram embeddings (Mikolov et al., 2013a;b) using a word2vec model trained on the training set. We use a vocabulary size of 7358 for words, and 2996 for entities. Gradients with respect to the coverage loss (Eq. 9) are only backpropagated in steps where no entity is annotated as being selected. To account for the false negatives in the training data due to the heuristic generation of the labels, gradients with respect to the entity selection loss are zeroed when no entity label is present.

### A.2   GENERATION MODEL

The hidden size of the context encoder is 200. The hidden size of the state vector encoder is 200. State vectors have dimensionality 30 (the same as in the neural process network). Dropout of 0.3 is used during training in the decoder. The context and state representations are projected jointly using an element-wise product followed by a linear projection Kim et al. (2016). Both encoders and the decoder are single layer. The learning rate is 0.0003 initially and is halved every 5 epochs. The model is trained with the Adam optimizer.

## B   TRAINING DETAILS OF BASELINES

### B.1   TRACKING BASELINES

**Joint Gated Recurrent Unit**   The hidden state of the GRU is 100. We use a dropout with a rate of 0.3 before any non-recurrent fully connected layers. We use the Adam optimizer with a learning rate of .001 and decay by a factor of 0.1 if we see no improvement on validation loss over a single epoch. We stop training early if the development loss does not decrease for five epochs. The batch size is 64. We use encoders, one for the entity selector, and one for the state change predictors. Word embeddings are initialized with skipgram embeddings using a word2vec model trained on the training set. We use a vocabulary size of 7358 for words.

**Recurrent Entity Networks**   Memory cells are tied to the entities in the document. For a recipe with 12 ingredients, 12 entity cells are initialized. All hyperparameters are the same as the in the bAbI task from Henaff et al. (2016). The learning rate start at 0.01 and is halved every 25 epochs. Entity cells and word embeddings are 100 dimensional. The encoder is a multiplicative mask initialized the same as in Henaff et al. (2016). Intermediate supervision from the weak labels is provided to help predict entities. A separate encoder is used for computing the attention over memory cells and the content to write to the memory. Dropout of 0.3 is used in the encoders. The batch size is 64. We use a vocabulary size of 7358 for words, and 2996 for entities.

### B.2   GENERATION BASELINES

**Seq2seq**   The encoder and decoder are both single-layer GRUs with hidden size 200. We use dropout with probability 0.3 in the decoder. We train with the Adam optimizer starting with a learning rate 0.0003 that is halved every 5 epochs. The encoder is bidirectional. The model is trained to minimize the negative loglikelihood of predicting the next word.

**Attentive Seq2seq**   The encoder is the same as in the seq2seq baseline. A multiplicative attention between the decoder hidden state and the context vectors is used to compute the attention over the context at every decoder time step. The model is trained with the same learning rate, learning schedule and loss function as the seq2seq baseline.

**EntNet Generator**   The model is trained in the same way as the NPN generator model in Appendix A.2 except that the state representations used as input are produced from by EntNet baseline described in Section 4.1 and Appendix B.1.

## C   ANNOTATIONS

### C.1   ANNOTATING STATE CHANGES

We provide workers with a verb, its definition, an illustrative image of the action, and a set of sentences where the verb is mentioned. Workers are provided a checklist of the six state change types and instructed to identify which of them the verb causes. They are free to identify multiple changes. Seven workers annotate each verb and we assign a state change based on majority vote. Of the set of 384 verbs extracted, only 342 have a state change type identified with them. Of those, 74 entail multiple state change types.

### C.2   ANNOTATING END STATES

We give workers a verb, a state change type, and an example with the verb and ask them to provide an end state for the ingredient the verb is applied to in the example. We then use the answers to manually aggregate a set of end states for each state change type. These end states are used as labels when the model predicting state changes. For example, a LOCATION change might lead to an end state of "pan," "pot", or "oven." End states for each state change type are provided in Table 7.

| State Change Type | End States |
|---|---|
| Temperature | hot; cold; room |
| Composition | composed; not composed |
| Cleanliness | clean; dirty; dry |
| Cookedness | cooked; raw |
| Shape | molded; hit; deformed; separated |
| Location | pan, pot, cupboard, screen, scale, garbage, 260 more |

Table 7: End states for each state change type

### C.3   ANNOTATING DEVELOPMENT AND TEST SETS

Annotators are instructed to note any entities that undergo one of the six state changes in each step, as well as to identify new combinations of ingredients that are created. For example, the sentence "Cut the tomatoes and add to the onions" would involve a SHAPE change for the tomatoes and a combination created from the "tomatoes" and "onions". In a separate task, three workers are asked to identify the actions performed in every sentence of the development and test set recipes. If an action receives a majority vote that it is performed, it is included in the annotations.

## D   ADDITIONAL RESULTS

### D.1   REMOVING TRAINING DATA

| Model | Entity Selection | | | State Change | |
|---|---|---|---|---|---|
| | F1 | UR | CR | F1 | ACC |
| 25% training data kept | 54.34 | 75.71 | 21.17 | 2.52 | 50.23 |
| 50% training data kept | 55.12 | 76.04 | 19.05 | 36.34 | 54.66 |
| 75% training data kept | 56.64 | 76.03 | 21.00 | 48.86 | 57.62 |
| 100% training data | 56.84 | 74.98 | 21.14 | 50.56 | 57.87 |

Table 8: Results for entity selection and state change selection on the development set when randomly dropping a percentage of the training labels

