# OpenReview forum: "Simulating Action Dynamics with Neural Process Networks"
_ICLR.cc/2018/Conference — Accept (Poster)_

### Official Review · AnonReviewer3 · 2017-11-25
**Worried about the generality of the model, the qualitative analysis, as well as a fair comparison to Recurrent Entity Networks and non-neural baselines**

**Rating:** 6
**Confidence:** 4

**Review:**

Summary

This paper presents Neural Process Networks, an architecture for capturing procedural knowledge stated in texts that makes use of a differentiable memory, a sentence and word attention mechanism, as well as learning action representations and their effect on entity representations. The architecture is tested for tracking entities in recipes, as well as generating the natural language description for the next step in a recipe. It is compared against a suit of baselines, such as GRUs, Recurrent Entity Networks, Seq2Seq and the Neural Checklist Model. While I liked the overall paper, I am worried about the generality of the model, the qualitative analysis, as well as a fair comparison to Recurrent Entity Networks and non-neural baselines.

Strengths

I believe the authors made a good effort in comparing against existing neural baselines (Recurrent Entity Networks, Neural Checklist Model) *for their task*. That said, it is unclear to me how generally applicable the method is and whether the comparison against Recurrent Entity Networks is fair (see Weaknesses).
I like the ablation study.

Weaknesses

While I find the Neural Process Networks architecture interesting and I acknowledge that it outperforms Recurrent Entity Networks for the presented tasks, after reading the paper it is not clear to me how generally applicable the architecture is. Some design choices seem rather tailored to the task at hand (manual collection of actions MTurk annotation in section 3.1) and I am wondering where else the authors see their method being applied given that the architecture relies on all entities and actions being known in advance. My understanding is that the architecture could be applied to bAbI and CBT (the two tasks used in the Recurrent Entity Networks paper). If that is the case, a fair comparison to Recurrent Entity Networks would have been to test against Recurrent Entity Networks on these tasks too. If they the architecture cannot be applied in these tasks, the authors should explain why.
I am not convinced by the qualitative analysis. Table 2 tells me that even for the best model the entity selection performance is rather unreliable (only 55.39% F1), yet all examples shown in Table 3 look really good, missing only the two entities oil (1) and sprinkles (3). This suggests that these examples were cherry-picked and I would like to see examples that are sampled randomly from the dev set. I have a similar concern regarding the generation task. First, it is not mentioned where the examples in Table 6 are taken from – is it the train, dev or test set? Second, the overall BLEU score seems quite low even for the best model, yet the examples in Table 6 look really good. In my opinion, a good qualitative analysis should also discuss failure cases. Since the BLEU score is so low here, you might also want to compare perplexity of the models.
The qualitative analysis in Table 5 is not convincing either. In Appendix A.1 it is mentioned that word embeddings are initialized from word2vec trained on the training set. My suspicion is that one would get the clustering in Table 4 already from those pretrained vectors, maybe even when pretrained on the Google news corpus. Hence, it is not clear what propagating gradients through the Neural Process Networks into the action embeddings adds, or put differently, why does it have to be a differentiable architecture when an NLP pipeline might be enough? This could easily be tested by another ablation where action embeddings are pretrained using word2vec and then fixed during training of the Neural Process Network. Moreover, in 3.3 it is mentioned that even the Action Selection is pretrained, which makes me wonder what is actually trained jointly in the architecture and what is not.
I think the difficulty of the task at hand needs to be discussed at some point, ideally early in the paper. Until examples on page 7 are shown, I did not have a sense for why a neural architecture is chosen. For example, in 2.3 it is mentioned that for "wash and cut" the two functions fwash and fcut need to be selected. For this example, this seems trivial as the functions have the same name (and you could even have a function per name!). As far as I understand, the point of the action selector is to only have a fixed number of learned actions and multiple words (cut, slice etc.) should select the same action fcut. Otherwise (if there is little language ambiguity) I would not see the need for a complex neural architecture. Related to that, a non-neural baseline for the entity selection task that in my opinion definitely needs to be added is extracting entities using a pretrained NER system and returning all of them as the selection.
p2 Footnote 1: So if I understand this correctly, this work builds upon a dataset of over 65k recipes from Kiddon et al. (2016), but only for 875 of those detailed annotations were created?

Minor Comments

p1: The statement "most natural language understanding algorithms do not have the capacity …" should be backed by reference.
p2: "context representation ht" – I would directly mention that this is a sentence encoding.
p3: 2.4: I have the impression what you are describing here is known in the literature as entity linking.
p3 Eq.3: Isn't c3*0 always a vector of zeros?
p4 Eq.6: W4 is an order-3 tensor, correct?
p4 Eq.8: What is YC and WC here and what are their dimensions? I am confused by the softmax, as my understanding (from reading the paragraph on the Action Selection Loss on p.5) was that the expression in the softmax here is a scalar (as it is done for every possible action), so this should be a sigmoid to allow for multiple actions to attain a probability of 1?
p5: "See Appendix for details" -> "see Appendix C for details"
p5 3.3: Could you elaborate on the heuristic for extracting verb mentions? Is only one verb mention per sentence extracted?
p5: "trained to minimize cross-entropy loss" -> "trained to minimize the cross-entropy loss"
p5 3.3: What is the global loss?
p6: "been read (§2.5." -> "been read (§2.5)."
p6: "We encode these vectors using a bidirectional GRU" – I think you composing a fixed-dimensional vector from the entity vectors? What's eI?
p7: For which statement is (Kim et al. 2016) the reference? Surely, they did not invent the Hadamard product.
p8: "Our model, in contrast" use" -> "Our model, in contrast, uses".
p8 Related Work: I think it is important to mention that existing architectures such as Memory Netwroks could, in principle, learn to track entities and devote part of their parameters to learn the effect of actions. What Neural Process Networks are providing is a strong inductive bias for tracking entities and learning the effect of actions that is useful for the task considered in this paper. As mentioned in the weaknesses, this might however come at the price of a less general model, which should be discussed.

# Update after the rebuttal
Thanks for the clarifications and updating the paper. I am increasing my score by two points and expect to see the ablations as well as the NER baseline mentioned in the rebuttal in the next revision of the paper. Furthermore, I encourage the authors to include the analysis of pretrained word2vec embeddings vs the embeddings learned by this architecture into the paper.

---

> ### Author Response · Authors · 2017-12-27
> **Re: Reviewer #3 part 1**
>
> Because our response was longer than 5000 characters, we separate our response into multiple parts to break the writing at natural breaks in the response.
>
> --- Motivation with Respect to Related Work ---
> We thank the reviewer for the question that prompts us to better clarify the key differences between previous approaches based on datasets such as bAbI, and the task proposed in our study.  The motivation of our work is to probe a research direction where we make use of naturally existing text with no gold labels, and investigate the role of the modular architecture and intermediate loss functions (with distant supervision) for learning latent action dynamics. In sum, the key contributions of our work are (1) to introduce a new task and dataset (including detailed annotations for evaluation) that bring up unique challenges that previous datasets did not cover, and (2) to propose a new model that is better suited for this new challenge of reasoning about action dynamics.
>
> As such, our newly introduced task actually complements work on densely labeled datasets such as bAbI. The bAbI dataset is synthetically constructed such that training labels cover the full spectrum of the semantics the model is expected to learn (i.e., # of training instances is extremely high for the # of words/concepts involved). When the training set provides sufficient inductive signals, it is possible to train an end-to-end model to extract the complex relations needed to do well on the task, and Recurrent Entity Networks are one of the best models architected. In our task, because the dataset alone does not provide sufficient inductive signals (only 875 recipes are densely labeled for evaluation), we investigate methods to provide better inductive biases using intermediate losses guided by distant supervision. We view both types of research directions --- integrating inductive biases into datasets (bAbI) vs. models (NPN) --- as important to pursue. They are complementary to each other, and our work focuses on the latter that has been relatively less explored in the existing literature.
>
> CBT is a cloze task based on children’s stories. While CBT is based on real natural language text like ours, the nature of the task differs much from ours in that answering the cloze task often requires remembering the surface patterns associated with each entity throughout the story excerpt (as has been also suggested by the Window Encoding used by prior approaches on this task). In contrast, our task focuses on the unspoken causal effects of actions, rather than explicitly mentioned descriptions about entities.
>
> Given the key differences between our task and others, it seems beyond the scope to require our model to outperform on all other tasks with different modeling requirements. That said, we are happy to include a detailed and insightful narrative about these differences in our revision, along with side by side performance comparisons. At this time, our conjecture is that updating entity states only through action application is likely to be too restrictive for CBT and bAbI tasks where remembering surface patterns without corresponding actions is crucial. However, our neural process networks can be easily extended to directly connect the sentence encoding to the simulation module (a minor change to one equation), in order to allow for updating entity representations even when there are no explicit actions associated.
>
> --- Qualitative Examples ---
> The examples in all tables were taken from the development set. We chose these examples to provide interesting case studies on some of the patterns that the model is able to learn by reading text and simulating the underlying world. We agree with the reviewer that an analysis of failure cases should have been included in the original submission and have updated the paper to include examples of similarly interesting cases the model misses. We intend to expand the model’s capabilities to capture these in future work.

---

> > ### Author Response · Authors · 2017-12-27
> > **Re: Reviewer #3 part 2**
> >
> >
> > --- Learning Causality-aware Action Embeddings ---
> > We apologize for the confusion about the action selector pretraining. What we meant in this case is that the MLP used to select action embeddings is pretrained. The action embeddings themselves are learned jointly with the entity selector, the simulation module, state change predictors and sentence encoder.
> >
> > We included Table 5 to show our action embeddings model semantic similarity between real-world actions. While word2vec embeddings would, no doubt, capture lexical semantics between these actions, the neural process network learns commonalities between actions that aren’t as extractable with a word2vec model. Looking at the action embeddings for ``bake”, ``boil”, and ``pour”, for example, we list the cosine similarities between pairs below:
> >
> > Skipgram:
> > boil - bake → 0.329
> > boil - pour → 0.548
> >
> > NPN:
> > boil - bake → 0.687
> > boil - pour → -0.119
> >
> > The NPN learns action embedding representations based on the state changes those actions induce as opposed to the local context windows around the mentions of the action in text, thereby encoding different semantics in the learned representation. While we did not use pretrained skipgram embeddings to initialize the action embeddings in our work, it is possible that including them when training our model might even lead to better results on our task as the action embeddings could encode elements of both lexical (word2vec) and frame (NPN) semantics. Conversely, we would argue that using only pretrained action embeddings from a word2vec model with no additional training would cause the bilinear matrix from the simulation module to have to learn the simulation mapping functions on its own, which would make the model less expressive. We will include both additional ablations in our final paper.
> >
> > For the moment, the action selector learns from distant supervision (string matching in each sentence is used to extract verb mention(s) as labels), but the model is designed to generalize beyond this signal. For example, in the sentence “Boil the water in the pot”, the model is designed to be able to select a composite action that includes an action such as f_put because boiling water involves moving the location of water to the pot. For the moment, we initialize a single action embedding for each verb in the lexicon and let the model learn to map sentences to a mask over these action embeddings. We agree with the reviewer, however, that it would be an interesting investigation to make the action embeddings ``implicit,” letting the model learn to select combinations of elementary actions. This approach is one of our current avenues of future work and could have the effect of generalizing the model similarly to the un-tied version of the REN.
> >
> > The reviewer makes a good point about including an NER baseline in the evaluation and we will include it in the final paper. We don’t anticipate the performance being much stronger than the GRU baseline, however, since current NER systems can only identify entities that are directly mentioned in the text, thereby missing elided, coreferent and composite mentions.

---

> > > ### Author Response · Authors · 2017-12-27
> > > **Re: Reviewer #3 part 3**
> > >
> > > --- Minor Comments ---
> > > We appreciate the reviewer’s thought-provoking questions about the impact of our model. We’ve updated the paper to extend the qualitative evaluation with additional examples and to clarify where our approach differentiates with the goals of general-purpose memory models such as Recurrent Entity Networks. We thank the reviewer for pointing out additional baselines and ablations to run to show the importance of the components of the model and will update the paper to incorporate them as we get the results. Finally, we appreciate the reviewer’s comments pointing out minor corrections to be made in the paper, and have incorporated them in the revised version.
> > >
> > > Below, we address minor comments made by the reviewer that were not addressed in the paragraphs above.
> > >
> > > p3: 2.4: I have the impression what you are describing here is known in the literature as entity linking.
> > >
> > > Assuming the reviewer is referring to the recurrent attention paragraph, we think coreference resolution would be a more accurate analogue to the task being handled as the goal of the recurrent attention mechanism is to tie connections between entity changes in the text without the use of an external KB. However, coreference tasks are defined only over explicitly mentioned entities in the text, while our task requires reasoning about implicit mentions as well, e.g., “Add water to the pot. Boil for 30 minutes” (where the implicit argument of Boil is water).
> > >
> > > p3 Eq.3: Isn't c3*0 always a vector of zeros?
> > >
> > > Yes, we included this option in the choice distribution as an easy short-circuit for the model to choose to include no entities in a particular step.
> > >
> > > p4 Eq.6: W4 is an order-3 tensor, correct?
> > >
> > > Yes, W_4 is a bilinear projection tensor between the action embedding and the entity embedding. We’ve clarified this in the new version of the paper
> > >
> > > p4 Eq.8: What is YC and WC here and what are their dimensions? I am confused by the softmax, as my understanding (from reading the paragraph on the Action Selection Loss on p.5) was that the expression in the softmax here is a scalar (as it is done for every possible action), so this should be a sigmoid to allow for multiple actions to attain a probability of 1?
> > >
> > > The softmax here predicts the end state for each state change in the lexicon. Each state change is predicted individually, so Y_c corresponds to the end state being predicted for an individual state change. W_c corresponds to a projection for each individual state change. Each state predictor is a separate multi-class classifier that predicts the end state of the entity from the output of the action applicator. These predictors are trained using the State Change loss in section 5. Actions are selected by the sigmoid in Equation 1.
> > >
> > > p6: "We encode these vectors using a bidirectional GRU" – I think you composing a fixed-dimensional vector from the entity vectors? What's eI?
> > >
> > > eI is the concatenation of final time step hidden states from encoding the entity state vectors in both directions using a bidirectional GRU.
> > >
> > > p7: For which statement is (Kim et al. 2016) the reference? Surely, they did not invent the Hadamard product.
> > >
> > > Kim et al. used the Hadamard product to jointly project two input representation in multimodal learning. We used their citation as a motivation for our decision to jointly project signal from the entity state vectors and word context representation. We’ve removed the citation to get rid of this ambiguity.

---

### Official Review · AnonReviewer1 · 2017-11-27
**good paper**

**Rating:** 9
**Confidence:** 4

**Review:**

SUMMARY.

The paper presents a novel approach to procedural language understanding.
The proposed model reads food recipes and updates the representation of the entities mentioned in the text in order to reflect the physical changes of the entities in the recipe.
The authors also propose a manually annotated dataset where each passage of a recipe is annotated with entities, actions performed over the entities, and the change in state of the entities after the action.
The authors tested their model on the proposed dataset and compared it with several baselines.


----------

OVERALL JUDGMENT
The paper is very well written and easy to read.
I enjoyed reading this paper, I found the proposed architecture very well thought for the proposed task.
I would have liked to see a little bit more of analysis on the results, it would be interesting to see what are the cases the model struggles the most.

I am wondering how the model would perform without intermediate losses i.e., entity selection loss and action selection loss.
It would also be interesting to see the impact of the amount of 'intermediate' supervision on the state change prediction.

The setup for generation is a bit unclear to me.
The authors mentioned to encode entity vectors with a biGRU, do the authors encode it in order of appearance in the text? would not it be better to encode the entities with some structure-agnostic model like Deep Sets?

---

> ### Author Response · Authors · 2017-12-27
> **Re: Reviewer #1**
>
> We thank the reviewer for their positive feedback. We share the same excitement about the potential for knowledge-guided architectures that simulate world dynamics.
>
> We’ve edited the paper to show more analysis examples. We’d originally shown examples that presented interesting case studies on the model’s capabilities. We’ve now added other interesting cases that the model fails to handle, but that future simulators would need to capture to correctly model the domain.
>
> --- Intermediate Losses for learning with Distant Supervision ---
> Thanks for the question about the impact of the intermediate modular loss as that was one of the key investigation points of our work: whether a neural network trained with a single loss (with distant supervision) could learn the internal dynamics of the task, or whether adding additional losses as guides (with additional distant supervision) would promote the architected inductive biases. This investigation point is a direct consequence of the fact that we do not assume a manually constructed dataset that provides sufficient annotated labels that support directly learning implied action dynamics. Instead, we make use of naturally existing data as is, and investigate the role of the modular architecture and distantly supervised intermediate losses for learning latent structure.
>
> To provide more detailed comments about the intermediate loss: without the entity selection and the action selection loss, the model would not learn the necessary bias to use the correct actions and entities in predicting the final states. Pretraining the action selector was also especially useful as it allowed the model to use the correct action embeddings when predicting the state changes that were happening in each step. This allowed errors in predicting the final states to be backpropagated to the correct action embeddings from the start.
>
> We also think it’s an interesting question to see how many examples the model must see during training to learn to select entities and simulate state changes. We thought about including experiments that randomly dropped a percentage of the training set labels and will add these ablations in the final paper.
>
> --- Generation Modeling Variations ---
> We appreciate the reviewer’s suggestion for using deep sets to encode the state vectors and agree that it seems like a better modeling fit at an intuitive level. While we did not try deep sets as an encoding method, in our pilot study, we explored several attention mechanisms over both the context words and the entity state vectors, and we found that the simple sequential encoding leads to the best performance, a conclusion that had also been found in prior work (Kiddon et al. 2016). We will look into deep sets as an encoding mechanism and report it in the final paper if helpful.

---

### Official Review · AnonReviewer2 · 2017-11-28
**Interesting model to incorporate domain-specific knowledge in procedural language understanding**

**Rating:** 8
**Confidence:** 4

**Review:**

The paper studies procedural language, which can be very useful in applications such as robotics or online customer support. The system is designed to model knowledge of the procedural task using actions and their effect on entities. The proposed solution incorporates a structured representation of domain-specific knowledge that appears to improve performance in two evaluated tasks: tracking entities as the procedure evolves, and generating sentences to complete a procedure. The method is interesting and presents a good amount of evidence that it works, compared to relevant baseline solutions.

The proposed tasks of tracking entities and generating sentences are also interesting given the procedural context, and the authors introduce a new dataset with dense annotations for evaluating this task. Learning happens in a weakly supervised manner, which is very interesting too, indicating that the model introduces the right bias to produce better results.

The manual selection and curation of entities for the domain are reasonable assumptions, but may also limit the applicability or generality from the learning perspective. This selection may also explain part of the better performance, as the right bias is not just in the model, but in the construction of the "ontologies" to make it work.

---

> ### Author Response · Authors · 2017-12-27
> **Re: Reviewer #2**
>
>
> --- Architectures with modular prior knowledge representations ---
> It is correct that our method assumes that predefined sets of entities, actions, and their causal effects are given before initializing the simulation environment. One motivation behind this design choice is to investigate more explicit and modular representations of the world (i.e., entities, actions, and their causal effects), abstracting away from specific words that appeared in the input text. We postulated that this modular architecture would better support integration of prior knowledge about actions and their causal effects, which can be viewed as part of the common sense knowledge people start with that biases how they read and interpret text. We agree with the reviewer that an interesting future research direction would be fully automatic acquisition of ontological knowledge, which we felt was beyond the scope of this paper.
>
> --- Mostly automatic acquisition of prior knowledge ---
> We also wonder whether there might have been slight confusion about how we acquire the predefined sets of entities, actions, and their causal effects. Importantly, for the training set, we acquire entities and actions automatically from the training corpus. We manually annotated entities and actions only for the purpose of evaluation, but do not use them during training. It is correct that we manually curate the handful of dimensions of action causality, however, primarily because there does not seem to be an easy way to acquire them automatically.

---

### Decision · Program_Chairs · 2018-01-29
**ICLR 2018 Conference Acceptance Decision**

**Decision:**

Accept (Poster)

**Comment:**

this submission proposes a novel extension of existing recurrent networks that focus on capturing long-term dependencies via tracking entities/their statesand tested it on a new task. there's a concern that the proposed approach is heavily engineered toward the proposed task and may not be applicable to other tasks, which i fully agree with. i however find the proposed approach and the authors' justification to be thorough enough, and for now, recommend it to be accepted.